# Contribution and Interaction of Shiga Toxin Genes to *Escherichia coli* O157:H7 Virulence

**DOI:** 10.3390/toxins11100607

**Published:** 2019-10-18

**Authors:** Gillian A.M. Tarr, Taryn Stokowski, Smriti Shringi, Phillip I. Tarr, Stephen B. Freedman, Hanna N. Oltean, Peter M. Rabinowitz, Linda Chui

**Affiliations:** 1Department of Pediatrics, Cumming School of Medicine, University of Calgary, Calgary, AB T3B 6A8, Canada; 2Department of Laboratory Medicine and Pathology, University of Alberta and Alberta Public Labs, Edmonton, AB T6G 2J2, Canada; 3Department of Veterinary Microbiology and Pathology, Washington State University, Pullman, WA 99163, USA; 4Division of Gastroenterology, Hepatology, and Nutrition, Washington University School of Medicine, St. Louis, MO 63110, USA; 5Washington State Department of Health, Shoreline, WA 98155, USA; 6Department of Environmental and Occupational Health Sciences, University of Washington, Seattle, WA 98195, USA

**Keywords:** *Escherichia coli* O157:H7, Shiga toxin-producing *Escherichia coli*, *stx* genes, hemolytic uremic syndrome

## Abstract

*Escherichia coli* O157:H7 is the predominant cause of diarrhea-associated hemolytic uremic syndrome (HUS) worldwide. Its cardinal virulence traits are Shiga toxins, which are encoded by *stx* genes, the most common of which are *stx1a*, *stx2a*, and *stx2c.* The toxins these genes encode differ in their in vitro and experimental phenotypes, but the human population-level impact of these differences is poorly understood. Using Shiga toxin-encoding bacteriophage insertion typing and real-time polymerase chain reaction, we genotyped isolates from 936 *E. coli* O157:H7 cases and verified HUS status via chart review. We compared the HUS risk between isolates with *stx2a* and those with *stx2a* and another gene and estimated additive interaction of the *stx* genes. Adjusted for age and symptoms, the HUS incidence of *E. coli* O157:H7 containing *stx2a* alone was 4.4% greater (95% confidence interval (CI) −0.3%, 9.1%) than when it occurred with *stx1a*. When *stx1a* and *stx2a* occur together, the risk of HUS was 27.1% lower (95% CI −87.8%, −2.3%) than would be expected if interaction were not present. At the population level, temporal or geographic shifts toward these genotypes should be monitored, and *stx* genotype may be an important consideration in clinically predicting HUS among *E. coli* O157:H7 cases.

## 1. Introduction

*Escherichia coli* O157:H7 is a leading cause of hospitalization for foodborne illness and the predominant cause of post-diarrheal hemolytic uremic syndrome (HUS) [1]. Characterized by hemolytic anemia, thrombocytopenia, and renal injury, HUS often necessitates renal replacement therapy, has a 1–5% case fatality [2,3,4], and is believed to be the consequence of vascular injury from circulating Shiga toxins (Stx) produced by this pathogen [5]. Children <5 years old suffer the highest incidence of reported *E. coli* O157:H7 infections, HUS, and death.

Pathogen characteristics are important factors in determining progression to HUS in humans infected with *E. coli* O157:H7 and other Stx-producing *E. coli* (STEC). STEC can express two families of Stx, its cardinal virulence factor, namely Stx1 and Stx2. These toxins are encoded by several allelic variants, of which *stx1a*, *stx2a*, and *stx2c* are most common among *E. coli* O157:H7 strains isolated from humans. The genotype of a single bacterial isolate may contain any or all of these subtypes. While Stx2, particularly when encoded by *stx2a* subtype, has been observed more frequently among cases with severe disease [6,7,8,9,10,11], previous studies have failed to estimate the risk of HUS attributable to particular genotypes or *stx* subtypes. Knowledge of HUS attributable risk by genotype is necessary to better understand patient prognosis. Moreover, STEC possessing *stx2a* as the sole Stx-encoding gene are isolated disproportionately from HUS cases, relative to STEC containing *stx1a* and *stx2a* [10,11,12,13,14], suggesting negative interaction between subtypes. This is a paradox, as one would expect that an *E. coli* O157:H7 that produces two Stx (i.e., Stx1 and Stx2a) would be more virulent than an *E. coli* O157:H7 producing Stx2a as the sole toxin.

To elucidate the role of *E. coli* O157:H7 *stx* genotypes in the development of HUS at the population level, we estimated (1) the risk of HUS associated with observed *stx* genotypes, (2) overall and age-specific changes in risk when *stx2a* is found alone vs. in combination with other *stx* alleles, and (3) the additive interaction between different *stx* alleles.

## 2. Results

We *stx*-genotyped 966 (83%) of the 1160 *E. coli* O157:H7 isolates from cases reported during the study period. We could not verify HUS status for 28 hospitalized cases, because they had no chart available for abstraction. Additionally, two of the isolates we attempted to genotype yielded no *stx* genes, leaving 936 cases for analysis of HUS risk (Appendix A). In total, HUS occurred in 69 cases of all ages (7.4%) (Table 1), including 52 of 372 (14.0%) children <10 years old (Appendix A).

*stx1a2a* was the most common genotype (439/936; 46.9%), followed by *stx2a*-only (225/936; 24.0%) and *stx2a2c* (190/936; 20.3%) genotypes (Table 1). Four other genotypes were isolated from the remaining 82 cases (8.8%); these were not included in the primary analysis. We observed a broad age distribution across all genotypes. The *stx2a*-only genotype was more commonly isolated from outbreak cases than other genotypes were. Diarrhea occurred in >98% of cases, regardless of genotype. Blood in the stool was slightly more common and vomiting was slightly less common among *stx1a2a* genotype cases, relative to *stx2a*-only and *stx2a2c* genotypes. Cases from which one of the less common genotypes was isolated experienced generally milder illnesses.

### 2.1. Risk of HUS

The cumulative incidence of HUS was greatest among *stx2a*-only-infected cases, with 10.7% [95% confidence interval (CI) 7.0%, 15.5%] of cases progressing to HUS (Figure 1). Cumulative incidence was 6.4% (95% CI 4.3%, 9.1%) among those with a *stx1a2a* strain. After adjusting for age and symptoms, HUS incidence of the *stx2a*-only genotype was 4.4% (95% CI −0.3%, 9.1%) greater than the *stx1a2a* genotype (Table 2). This means that if the 439 individuals infected with *stx1a2a* had instead been infected with an *stx2a*-only strain, 47 cases of HUS would have been expected to occur, or 19 more than the 28 cases of HUS observed, age and symptoms being equal. The adjusted relative risk (RR) was 1.58 (95% CI 0.98, 2.56).

HUS incidence was 8.4% (95% CI 4.9%, 13.3%) among *stx2a2c*-infected cases (Figure 1). There was no difference between the *stx2a*-only and *stx2a2c* genotypes (adjusted risk difference (RD) 3.0%; 95% CI −2.6%, 8.7%). Similarly, the adjusted RR was 1.43 (95% CI 0.83, 2.45) (Table 2).

The only other genotype in which HUS occurred was *stx1a*-only, in a single case (incidence 8.3%; 95% CI 0.2%, 38.5%) (Appendix A). For the “other” genotypes, the aggregate HUS incidence was 1.2% (95% CI 0.03%, 6.6%) (Figure 1). Adjusted for age and symptoms, HUS incidence of the *stx2a*-only genotype was 5.8% (95% CI 1.3%, 10.3%) greater than the other genotypes combined (Table 2). The adjusted RR was 4.65 (95% CI 0.70, 31.08).

### 2.2. Risk of HUS in Children

Children <10 years constituted 40% (372/936) of reported *E. coli* O157:H7 cases but 75% (52/69) of HUS cases (Appendix A). In children <5 years old, the cumulative incidence of HUS ranged from 13.8 to 16.9% for the three primary genotypes; in children 5–9 years old, it ranged from 9.2 to 17.5%. When limiting the sample to children <10 years, crude RD point estimates were similar to those obtained in the full sample (e.g., 4.1% for *stx2a* vs. *stx1a2a*); however, CIs overlapped 0 by a wide margin (e.g., −4.8%, 13.0% for *stx2a* vs. *stx1a2a*), and there were too few events to have confidence in the adjusted models (Appendix A).

### 2.3. Risk of Renal Replacement Therapy

The cumulative incidence of renal replacement therapy (RRT) was 6.2% (95% CI 3.4%, 10.2%) among *stx2a*-only cases, 2.5% (95% CI 1.3%, 4.4%) among *stx1a2a* cases, 5.3% (95% CI 2.6%, 9.5%) among *stx2a2c* cases, and 1.2% (95% CI 0.03%, 6.5%) among cases infected with strains with other *stx* genotypes. After adjusting for age and symptoms, RRT incidence of the *stx2a*-only genotype was 3.8% (95% CI 0.2%, 7.4%) greater than the *stx1a2a* genotype. We did not detect a conclusive difference between the *stx2a*-only and *stx2a2c* (RD 1.4%; 95% CI −3.3%, 6.1%) or between the *stx2a*-only and other genotypes (RD 2.1%; 95% CI −1.6%, 5.9%).

### 2.4. stx Allelic Interaction

We detected sub-additive interaction of *stx1a* and *stx2a*. After accounting for age and symptoms, when *stx1a* and *stx2a* occur together in *E. coli* O157:H7 (i.e., as the *stx1a2a* genotype), the risk of HUS was 27.1% lower (95% CI −87.8%, −2.3%) than would be expected if interaction were not present, based on the risk of HUS associated with the *stx1a*-only and *stx2a*-only genotypes (Appendix A). No interaction was observed for *stx2a* and *stx2c* (−3.0%; 95% CI −8.5%, 2.4%).

### 2.5. Sensitivity Analysis

After restricting the analysis to hospitalized cases and adjusting for age, fever, and antibiotic use, RD and RR point estimates slightly exceeded or were similar to those in the primary analysis (Appendix A). For *stx2a*-only vs. *stx1a2a*, the RD was 7.2% with a wide confidence interval (95% CI −2.2%, 16.5%), and the RR was 1.34 (95% CI 0.87, 2.05).

After reassigning the genotype of the isolates that might have lost an *stx* gene after isolation, the excess risk associated with the *stx2a*-only genotype was similar or marginally in excess of that seen in the primary analysis. The adjusted RD of HUS incidence of *stx2a*-only vs. *stx1a2a* was 5.2% (95% CI 0%, 10.4%), and the adjusted RR was 1.7 (95% CI 1.04, 2.79) (Appendix A).

## 3. Discussion

We have demonstrated that the risk of HUS is substantially associated with the *E. coli* O157:H7 *stx* genotype. For the major genotypes, cumulative incidence of HUS increased from 6.4% for the *stx1a2a* genotype to 10.7% for the *stx2a*-only genotype. Among children <5 years, the incidence of HUS increased from 13.8% for the *stx2a2c* genotype to 16.9% for the *stx2a*-only genotype. We estimated that the *stx2a*-only genotype causes HUS in 4.4% more infected cases (i.e., ~4 more HUS cases per 100 *E. coli* O157:H7 cases) than the *stx1a2a* genotype does, adjusting for age and symptoms. Moreover, we found a strong negative interaction on the additive scale between *stx1a* and *stx2a*. Similarly, the *stx2a*-only genotype increased risk of RRT by 3.8%. We did not identify a difference in risk between the *stx2a*-only and *stx2a2c* genotypes.

Since early *E. coli* O157:H7 outbreaks, epidemiologic studies have found an association between Stx2 and HUS. However, with few exceptions [6,9,11,15], studies have not offered genotype-specific incidence or adjusted measures of excess risk. Consequently, the magnitude of how genotypic variations in virulence impact HUS incidence across populations has been unclear. We identified a substantial increase in HUS risk associated with the *stx2a* genotype as compared to the *stx1a2a* genotype, two of the most common genotypes in North America and Japan [9,11,16,17,18,19]. The virulence of the *stx2a2c* genotype, predominant in other settings [7,8,20,21], was similar to that of the *stx2a*-only genotype. This increased risk is not trivial. Over our 10-year study period in Washington State, we would have expected to see 19 additional HUS cases had *stx2a* strains replaced *stx1a2a* strains, increasing the HUS rate by a quarter, all else being equal. On a relative scale, that would amount to a 28% increase in the number of HUS cases. The public health impact of shifts in the *E. coli* O157 population is not merely theoretical. A relative decline in *stx1a2a* infections and increase in *stx2a*, with or without *stx2c*, can be observed between early studies in Washington State [11] and our current data, and we have shown a similar secular bacterial population shift on the absolute scale within the 10 years of our study [22]. These differences may explain variation in HUS incidence between geographic regions where the dominant *stx* genotypes differ.

Several lines of experimental evidence are harmonious with our epidemiologic findings that *E. coli* O157:H7 containing one toxin gene (i.e., *stx2*) are more virulent than *E. coli* O157:H7 containing two toxin genes (i.e., *stx1stx2*). Donohue-Rolfe et al. increased the neurovirulence of an *stx2*+/*stx1*+ *E. coli* O157:H7 in a gnotobiotic piglet model by deleting the *stx1* gene [13], Russo et al. showed that enterally administered Stx1a reduces enterally-administered Stx2a-mediated toxicity [23], and Petro et al. reported that Stx1a neutralizes the toxicity of Stx2 [14]. This last paper postulated that the effect is probably caused by competitive inhibition of the more potent Stx2a by the less potent Stx1a, and we favor this interpretation. However, they also note that they could not exclude the possibility that the attenuation of Stx2a toxicity is related to a less specific immunomodulatory effect of Stx1a, which diminishes host cell response to Stx2a intoxication. The exact mechanism underlying this paradoxical genotype finding is likely to remain speculative for some time. Nonetheless, our multivariable analysis of bacterial genotype and disease outcome resembles findings from smaller, older studies in which only univariate associations were tested, lending human relevance to the experimental data.

We have previously reported lack of association between phylogenetic lineages and HUS in this population, both overall and among children <10 years of age [24]. There is some concordance between lineage and *stx* genotype, with the most common human-biased lineages each having a dominant genotype [24]. However, genotypic variation within lineages does exist. That we observed an association between *stx* genotype and HUS where we failed to observe one between phylogenetic lineage and HUS suggests that lineage is an imperfect surrogate for *stx* genotype.

When stratifying by age group, we observed a slightly lower point estimate among children <10 years as among all cases when comparing *stx2a*-only and *stx1a2a* genotypes, but the estimate was imprecise. Although this may indicate there is no association between *stx* genotype and HUS in this age group, it is more likely that our sample size was insufficient to detect the difference with precision. Although phylogenetic lineage analysis previously suggested the *stx1a2a* genotype conferred protection to children <5 years [24], we did not observe this reversal in the current study. However, given the imprecision, we cannot say conclusively that the association holds in children <10 years. We believe further study is warranted to understand the effect of genotype in specific age groups.

The *stx2a*-only genotype is also more associated with greater HUS incidence than the less common ‘other’ genotypes included in our study. Notably, only 6 of these 82 isolates possessed a *stx2a* gene, and those were combined with *stx1a* (as well as *stx2c* in some), so we would expect their virulence to be attenuated relative to *stx2a*-only isolates. This is supported by HUS frequencies by genotype in some other studies [6,8,21], but not all [7]. Previous work has shown that many of these specific isolates are in cattle-biased phylogenetic lineages [24] that are rarely isolated from humans. Factors that confer lower infectivity and/or pathogenicity may also be involved in their lower virulence.

Only one case in our study developed HUS among those infected by one of the ‘other’ genotypes. This case was an otherwise healthy young child with no unusual risk factors. The isolate from this case was previously typed into the phylogenetic lineage dominated by the *stx1a2a* genotype. It is possible that a *stx2a* allele was lost in the host or on plating, and the Stx-encoding bacteriophage insertion (SBI) type of this specific isolate, and three others included in this analysis, is consistent with a *stx1a2a* isolate that lost its *stx2a* bacteriophage. Loss of *stx* genes from cultured isolates is a recognized phenomenon, with all or only a subset of genes being lost [12,25,26,27,28], and in broth culture, a subset of *E. coli* O157:H7 underwent spontaneous excision of the bacteriophages containing the *stx* genes [25]. We excluded two isolates that had no *stx* genes and explored the possibility that other isolates had lost one or more *stx* genes. When using lineage typing and pulsed field gel electrophoresis (PFGE) patterns as a guide to reassign the genotype of isolates that had potentially lost a gene, the RD between the *stx2a*-only and *stx1a2a* genotypes increased and HUS risk associated with the *stx2a*-only and *stx2a2c* genotypes was similar, supporting our primary findings.

While our study focused on *E. coli* O157:H7 cases, our findings may be more broadly applicable to other STEC given that Stx toxicity is driven largely by the B subunit of the toxin [29]. Non-O157 STEC are increasingly reported because of the widespread adoption of multiplex PCR platforms that detect *stx1* and *stx2* sequences. However, caution should be exercised when detecting a *stx1a*-only *E. coli* O157:H7 because of the potential that *stx2a* was lost or because of sampling error, *stx1a*-only non-O157 STEC are more common and thus more likely to be genuine results. However, not all multiplex panels distinguish *stx1* and *stx2*, and some laboratories choose not to release *stx* type; in these cases, follow-up enzyme immunoassays for the toxin, PCR, or whole genome sequencing should be considered if knowledge of the *stx* genotype would contribute to patient care.

Our data have implications for etiologic studies, aside from outbreak analyses, and intervention studies designed to prevent *E. coli* O157:H7 infections from progressing to HUS. Knowledge of genotype can be used to evaluate effect heterogeneity and could potentially confound apparent associations. In future small clinical trials, stratified randomization by genotype should be considered, if feasible. If not, the infecting strain genotype should be taken into account in the analysis of efficacy of the intervention. Additionally, *stx* genotype may be a candidate for clinical decision support tools designed to predict the development of HUS and need for RRT.

The analysis was restricted to those cases who sought care, provided a specimen, and were reported. People infected with *E. coli* O157 who do not seek care presumably have less severe disease. Factors driving care-seeking and stool testing include the presence of hematochezia, pain, a high number of diarrheal episodes per day, or concomitant vomiting. If certain genotypes are less likely to cause these symptoms, their proportions of HUS will be overestimated relative to genotypes more likely to cause these symptoms. If the *stx2a*-only genotype is more likely to cause severe symptoms than the referent genotype (e.g., *stx1a2a*), the referent genotype is likely underreported to a greater degree than *stx2a*. Assuming non-severe *E. coli* O157:H7 cases who were not reported did not have HUS, the true RD and RR would be greater than that observed in this case. Conversely, if the referent genotype causes more severe symptoms than the *stx2a*-only genotype, the true RD and RR would be lower than those we observed.

Our analysis was limited by missing genotype and HUS status. The largest factor influencing isolate missingness was year of isolation. The most probable associated mechanisms, such as genomic degradation and being misplaced, are likely to be random and not related to genotype or HUS status. Thus, we believe that our analysis is unbiased by these gaps in data. Moreover, among genotyped isolates, those missing HUS status were proportionally distributed across genotypes, suggesting there would be little change in the estimates. The association between fever and missing HUS status likely reflects that patients are more likely to seek care if they have a fever but does not impact the association between genotype and HUS.

## 4. Conclusions

We have estimated the excess risk of HUS and RRT attributable to *stx2a*-only *E. coli* O157:H7, relative to the *stx1a2a* genotype, demonstrating the population-level implications of the purported attenuation of *stx2a* virulence by *stx1a.* If *stx2a*-only strains had replaced *stx1a2a* strains in our population, we would have observed >25% more HUS cases. Public health officials should be aware of temporal and geographic variation in dominant *stx* genotypes, because they may imply an escalation of the risk of HUS among infected individuals. While the risk of HUS among *stx1a2a*-infected cases is less than if they were infected with a *stx2a*-only isolate, these patients are still at risk of HUS and should be managed with that possibility in mind. Future work is needed to understand genotypic HUS risk in different age groups and acuity levels.

## 5. Materials and Methods

### 5.1. Study Population

We conducted a population-based retrospective cohort study of all culture-confirmed *E. coli* O157:H7 cases reported to the Washington State department of health (DOH) from 2005 through 2014. Demographic information, potential exposures, and details of the course of illness were obtained from case report forms. We confirmed HUS status, the primary outcome, and RRT, the secondary outcome, during a review of hospitalized *E. coli* O157:H7 cases from the study sample. Data collection has been detailed previously [24]. HUS was defined as hematocrit <30%, platelet count <150,000/mm^3^, and serum creatinine concentration above the normal for age [30], with all criteria met on the same day. We assumed all non-hospitalized cases did not have HUS because of the severity of this disease outcome.

The Washington State institutional review board designated this study as exempt. The University of Calgary conjoint health research ethics board approved this study.

### 5.2. Genotyping

*stx* genotype was the exposure of interest, classified as combinations of *stx1a, stx2a,* and *stx2c*. The three most common genotypes are *stx1a*-/*stx2a*+/*stx2c*-, *stx1a*+/*stx2a*+/*stx2c*-, and *stx1a*-/*stx2a*+/*stx2c*+. We refer to them as *stx2a*-only, *stx1a2a*, and *stx2a2c*, respectively, for the purpose of this paper. Isolates were genotyped in two batches.

SBI typing, a multiplex PCR method, was used to characterize the genotypes of 690 isolates at Washington State University. The SBI typing detects 12 targets identifying the insertion of three Stx-encoding bacteriophages and three specific *stx* subtypes (*stx1a*, *stx2a*, and *stx2c*) and has been described previously [31,32].

An additional 276 isolates were genotyped at the University of Alberta using a real-time PCR assay, described previously [33,34]. Isolates were grown on sheep blood agar plates (Dalynn Biologicals, Calgary, AB, Canada) for 24 h. For DNA extraction, a colony from each plate was mixed with rapid lysis buffer (100 mmol/L NaCl, 10 mmol/L Tris-HCl. pH 8.3, 1 mmol/L EDTA, pH 9.0; 1% Triton X-100) and then heated to 95 °C for 15 min, followed by centrifugation at 13,000g for 15 min. Real-time PCR was performed on the supernatant as DNA template using either a hydrolysis probe or SYBR green based approach. For the probe based assays the total reaction volume (20 μL) consisted of: 10 μL PrimeTime® Gene Expression Master Mix (Integrated DNA Technologies, Coralville, IA, USA), 0.9 μM of each primer, 0.25 μM of each probe, 5 μL of DNA template, and nuclease-free water (Invitrogen, Carlsbad, CA, USA). As for the SYBR green based assays, the total reaction volume (20 μL) consisted of: 10 μL Fast SYBR Green Master Mix (Applied Biosystems, Foster City, CA, USA), 0.3 μM of each primer, 5 μL of DNA template, and nuclease-free water. All reactions were performed using primers and probes ordered from Integrated DNA Technologies (Coralville, IA, USA), and were run on Applied Biosystems® 7500 fast Real-Time PCR system (Applied Biosystems, Foster City, CA, USA).

### 5.3. Statistical Analysis

Cases were excluded from analysis if the associated *E. coli* O157:H7 isolate was not genotyped, no *stx* genes were identified during typing, or if the case was hospitalized but HUS status could not be verified. We summarized demographic and illness characteristics with descriptive statistics and investigated potential reasons for missing specimens.

We calculated the cumulative incidence and exact binomial 95% CI of HUS by *stx* genotype as the number of cases infected with *E. coli* O157:H7 of a given genotype who developed HUS over the total number of cases infected with *E. coli* O157:H7 of that genotype. We assumed the incidence of HUS among *E. coli* O157:H7-infected patients unexposed to *stx1a*, *stx2a*, and *stx2c* was the incidence of atypical HUS, an entirely unrelated disorder, in the general population, 2 × 10^−6^ [35], or essentially 0. As such, we interpreted cumulative incidence estimates as attributable risks.

To estimate the excess risk of HUS associated with *stx2a*-only *E. coli* O157:H7 isolates, we calculated the RD and RR of that genotype vs. *stx1a2a*, *stx2a2c*, and all other genotypes combined. We adjusted RD and RR estimates for age, to control for confounding, and symptoms most likely to influence clinical management, specifically blood in the stool, vomiting, and fever, to block an indirect causal pathway between *stx* genotype and HUS (Figure 2; Appendix A). We used linear regression with robust standard errors to estimate RDs and modified Poisson regression with robust standard errors to estimate RRs. We also calculated the RD for our secondary outcome RRT, fully adjusted for age and symptoms. In secondary analysis, we estimated the RD and RR for HUS, restricting the sample to children <10 years old.

We calculated the degree of additive interaction between *stx1a* and *stx2a*, and *stx2a* and *stx2c*, separately, using the equation $p_{11}-p_{10}-p_{01}+p_{00}\;$, where the subscripts indicate the presence or absence of a specific gene. Values >0 and <0 can be interpreted as super-additive and sub-additive interaction, respectively. Three-way interaction was not considered because of the rarity of *stx1a+/stx2a+/stx2c+* isolates. We adjusted for age and symptoms using stabilized inverse probability weights. We used a bootstrap with 10,000 replicates to calculate 95% bias-corrected and accelerated CIs for interaction estimates. Multiplicative interaction was not directly assessed (Appendix A).

#### Sensitivity Analysis

To more fully account for the indirect causal pathway between *stx* genotype and HUS involving symptoms and clinical management, we conducted a sensitivity analysis of RD and RR estimates including adjustment for antibiotic use. Antibiotic use was only validated among hospitalized patients, thus limiting this analysis to that sub-population.

Acknowledging the possibility that *E. coli* O157:H7 may lose their *stx* bacteriophages after they are plated on agar, we conducted a sensitivity analysis reclassifying isolates with atypical *stx* genotype for their PFGE pattern, confirmed using phylogenetic lineage typing [24] (Appendix A).

## Figures and Tables

**Figure 1 toxins-11-00607-f001:**
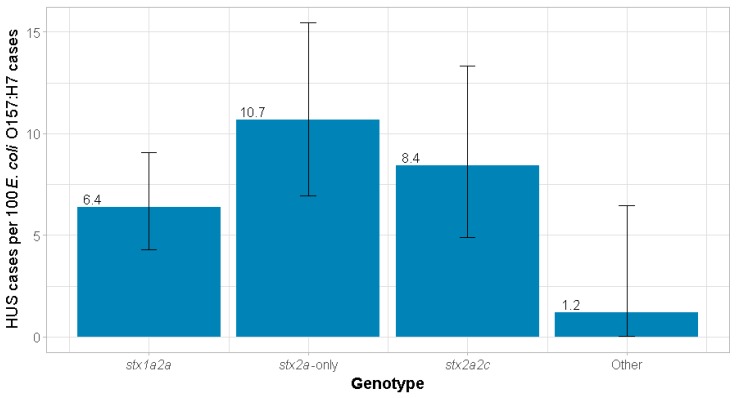
Cumulative incidence of HUS by *E. coli* O157:H7 genotype. Error bars represent 95% exact binomial confidence intervals. Abbreviation: HUS, hemolytic uremic syndrome.

**Figure 2 toxins-11-00607-f002:**
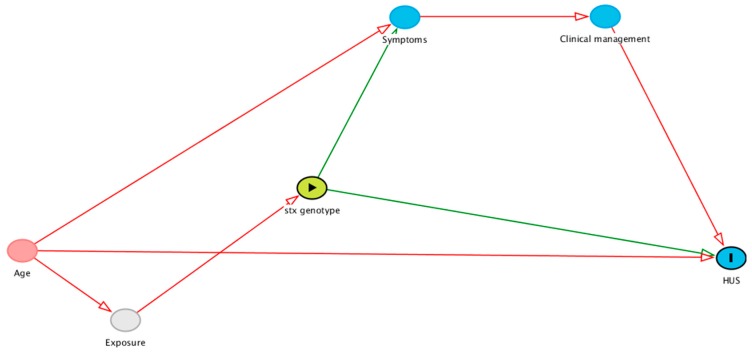
Directed acyclic graph of hypothesized relationships among *stx* genotype (exposure, green oval with triangle), HUS (outcome, blue oval with bar), and covariates. Age confounds the genotype-HUS relationship. There is a potential indirect pathway from genotype to HUS through symptoms (e.g., blood in stool, vomiting, fever) and clinical management (e.g., antibiotic use, intravenous fluid administration). Abbreviation: HUS, hemolytic uremic syndrome.

**Table 1 toxins-11-00607-t001:** Characteristics of *E. coli* O157 cases by Shiga toxin genotype.

	*stx1a2a* (*n* = 439)	*stx2a*-only (*n* = 225)	*stx2a2c* (*n* = 190)	Other (*n* = 82)	Overall (*n* = 936)
Age (years)					
Median (IQR)	18.0 (5.00, 42.5)	11.0 (4.00, 30.0)	12.0 (4.00, 33.0)	23.0 (8.00, 46.8)	16.0 (5.00, 39.3)
Comorbidity					
Present	50 (11.4%)	26 (11.6%)	18 (9.5%)	7 (8.5%)	101 (10.8%)
Absent	350 (79.7%)	184 (81.8%)	158 (83.2%)	69 (84.1%)	761 (81.3%)
Missing	39 (8.9%)	15 (6.7%)	14 (7.4%)	6 (7.3%)	74 (7.9%)
Outbreak-related					
Yes	42 (9.6%)	32 (14.2%)	14 (7.4%)	1 (1.2%)	89 (9.5%)
No	397 (90.4%)	193 (85.8%)	176 (92.6%)	81 (98.8%)	847 (90.5%)
Diarrhea					
Present	433 (98.6%)	221 (98.2%)	187 (98.4%)	81 (98.8%)	922 (98.5%)
Absent	2 (0.5%)	3 (1.3%)	1 (0.5%)	0 (0%)	6 (0.6%)
Missing	4 (0.9%)	1 (0.4%)	2 (1.1%)	1 (1.2%)	8 (0.9%)
Blood in stool					
Present	398 (90.7%)	189 (84.0%)	161 (84.7%)	61 (74.4%)	809 (86.4%)
Absent	31 (7.1%)	32 (14.2%)	22 (11.6%)	20 (24.4%)	105 (11.2%)
Missing	10 (2.3%)	4 (1.8%)	7 (3.7%)	1 (1.2%)	22 (2.4%)
Vomiting					
Present	204 (46.5%)	121 (53.8%)	102 (53.7%)	28 (34.1%)	455 (48.6%)
Absent	223 (50.8%)	103 (45.8%)	82 (43.2%)	52 (63.4%)	460 (49.1%)
Missing	12 (2.7%)	1 (0.4%)	6 (3.2%)	2 (2.4%)	21 (2.2%)
Abdominal pain					
Present	408 (92.9%)	204 (90.7%)	175 (92.1%)	71 (86.6%)	858 (91.7%)
Absent	16 (3.6%)	13 (5.8%)	8 (4.2%)	8 (9.8%)	45 (4.8%)
Missing	15 (3.4%)	8 (3.6%)	7 (3.7%)	3 (3.7%)	33 (3.5%)
Fever					
Present	167 (38.0%)	78 (34.7%)	70 (36.8%)	20 (24.4%)	335 (35.8%)
Absent	245 (55.8%)	132 (58.7%)	107 (56.3%)	53 (64.6%)	537 (57.4%)
Missing	27 (6.2%)	15 (6.7%)	13 (6.8%)	9 (11.0%)	64 (6.8%)
Hospitalized					
Yes	167 (38.0%)	93 (41.3%)	85 (44.7%)	20 (24.4%)	365 (39.0%)
No	263 (59.9%)	130 (57.8%)	105 (55.3%)	60 (73.2%)	558 (59.6%)
Missing	9 (2.1%)	2 (0.9%)	0 (0%)	2 (2.4%)	13 (1.4%)
HUS					
Yes	28 (6.4%)	24 (10.7%)	16 (8.4%)	1 (1.2%)	69 (7.4%)
No	411 (93.6%)	201 (89.3%)	174 (91.6%)	81 (98.8%)	867 (92.6%)
RRT					
Yes	11 (2.5%)	14 (6.2%)	10 (5.3%)	1 (1.2%)	36 (3.8%)
No	428 (97.5%)	211 (93.8%)	180 (94.7%)	81 (98.8%)	900 (96.2%)

Patient or caregiver reported presence of fever; if temperature was reported, fever was defined as ≥38.0 °C. ‘Other’ genotype includes *stx1a*-only, *stx1a2a2c*, *stx1a2c*, and *stx2c*-only genotypes. Abbreviations: HUS, hemolytic uremic syndrome; IQR, interquartile range; RRT, renal replacement therapy.

**Table 2 toxins-11-00607-t002:** Excess risk of HUS due to *stx2a* vs. other genotypes.

*stx2a* vs.	RD (95% CI)			RR (95% CI)		
	Crude	Age-Adjusted	Fully Adjusted	Crude	Age-Adjusted	Fully Adjusted
*stx1a2a*	0.043 (−0.003, 0.089)	0.036 (−0.010, 0.082)	0.044 (−0.003, 0.091)	1.67 (0.99, 2.82)	1.52 (0.9, 2.54)	1.58 (0.98, 2.56)
*stx2a2c*	0.022 (−0.034, 0.079)	0.021 (−0.035, 0.077)	0.03 (−0.026, 0.087)	1.27 (0.69, 2.31)	1.24 (0.68, 2.25)	1.43 (0.83, 2.45)
Other	0.095 (0.048, 0.141)	0.082 (0.037, 0.127)	0.058 (0.013, 0.103)	8.96 (1.23, 65.2)	7.28 (0.98, 54.14)	4.65 (0.7, 31.08)

Fully adjusted models are adjusted for age, blood in stool, vomiting, and fever. Abbreviations: CI, confidence interval; HUS, hemolytic uremic syndrome; RD, risk difference; RR, relative risk.

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
