# Peer review of "Contribution and Interaction of Shiga Toxin Genes to Escherichia coli O157:H7 Virulence"

_toxins, 2019, doi:10.3390/toxins11100607_

Round 1

Reviewer 1 Report

Dear Authors,

This is a well-written manuscript describing the correlation between the risk of developing Hemolytic Uremic Syndrom (HUS) and infection with Escherichia coli O157:H7 encoding different combinations of shiga toxin variants. The authors conclude that strains with the stx2a allele alone are associated with higher risk of developing HUS than strains with the stx1a and stx2a alleles and suggest that stx-typing can be an important diagnostic tool.

There are only a few minor concerns:

Lines 35, 37, 40, 41 and 43, the genes should be italicized.

The quality/sharpness of table 1 is poor. Gene names should be italicized.                                                                                                                            

Line 79, define RR the first time it occurs in the text.

The quality/sharpness of the text in figure 1 is poor.

Line 86, define RD the first time it occurs in the text.

Author Response

Lines 35, 37, 40, 41 and 43, the genes should be italicized.

Thank you for pointing these out. We have italicized all but those on line 43. The Stx1 and Stx2a on line 43 are the toxins, not the genes.

The quality/sharpness of table 1 is poor. Gene names should be italicized.

We have replaced Table 1 with an editable version, which is both clearer and appropriately italicized.

Line 79, define RR the first time it occurs in the text.

We have defined RR.

The quality/sharpness of the text in figure 1 is poor.

We have enlarged and increased the clarity of the text in Figure 1.

Line 86, define RD the first time it occurs in the text.

We have defined RD.

Reviewer 2 Report

This manuscript provides critical new information about the relationship between Shiga toxin gene expression and clinical outcomes. I am strongly supportive of the publication of the manuscript. I have only one suggestion - in the discussion, it will be beneficial if the authors provide a hypothesis about the mechanisms that may allow Stx1a production to attenuate toxicity of Stx2a only.

Author Response

This manuscript provides critical new information about the relationship between Shiga toxin gene expression and clinical outcomes. I am strongly supportive of the publication of the manuscript. I have only one suggestion - in the discussion, it will be beneficial if the authors provide a hypothesis about the mechanisms that may allow Stx1a production to attenuate toxicity of Stx2a only.

We appreciate this suggestion and have expanded our discussion of the existing experimental evidence, and, with it, a hypothesis about the mechanism at play. See page 6, lines 148-160.
